# The fate of *Candida tropicalis* in the black soldier fly larvae and its nutritional effect suggest indirect interactions

Lilach Ben-Mordechai[1,2], Neta Herman[1,2], Tzach Vitenberg[1], Sivan Margalit[2], Gianluca Tettamanti[3,4], Morena Casartelli[4,5], Daniele Bruno[3], Itai Opatovsky[1,2]*

1 Laboratory of Insect Nutrition and Metabolism, The Department of Nutrition and Natural Products, MIGAL-Galilee Research Institute, Kiryat Shmona, Israel, 2 Department of Animal Science, Faculty of Sciences and Technology, Tel-Hai Academic College, Upper Galilee, Israel, 3 Department of Biotechnology and Life Sciences, University of Insubria, Varese, Italy, 4 Interuniversity Center for Studies on Bioinspired Agro-environmental Technology (BAT Center), University of Napoli Federico II, Portici, Italy, 5 Department of Biosciences, University of Milano, Milano, Italy

* itaio@migal.org.il

## Abstract

Bacteria are known to colonize the insect gut and determine a positive effect on their host's fitness, for example, by providing essential nutrients or improving digestion efficiency. However, information on the colonization of the insect gut by fungi and their nutritional contribution is still scarce and fragmentary. In this study, the presence of *Candida tropicalis,* a fungus abundant in the black soldier fly (*Hermetia illucens,* BSF) larvae's gut and environment, was determined in the different gut regions. In addition, metabolites present in larvae fed with a fungus-containing diet were determined by untargeted metabolomics and compared to the *C. tropicalis* metabolic composition and metabolic changes in the feeding substrate supplemented with the microorganism. Our results indicate that *C. tropicalis* ceased to be present in the BSF gut after its supplementation in the feeding substrate was stopped, indicating that *C. tropicalis* does not colonize the gut. Larvae that were reared on diet supplemented with *C. tropicalis* displayed an increase in the fatty acid biosynthesis pathway, due to an increase in the palmitic and myristic acids that are abundant in *C. tropicalis*. The presence of *C. tropicalis* in the substrate caused an increase in threonine, leucine, and isoleucine biosynthesis pathways in the larvae and suggests indirect feeding from the fungal excretions in the substrate. In addition, the lysozyme activity in the larval gut was reduced by the presence of *C. tropicalis*, suggesting the fungal involvement in the digestive process for increasing fungal survival. This study suggests indirect symbiotic interactions, in which *C. tropicalis* thrives in the BSF larvae's environment and manipulates BSF digestive enzyme production to survive in this environment, but on the other hand, BSF larvae benefit metabolically from the *C. tropicalis* presence in its surrounding environment.

**Data availability statement:** All data files are available in the Zenodo database (URL https://zenodo.org/records/14878524).

**Funding:** Israel Science Foundation (grant No. 1167/21).

**Competing interests:** The authors have declared that no competing interests exist.

## Introduction

Interactions between insects and their gut microorganisms, as well as their role and significance, are relatively well known [1–3]. These relationships can be positive and defined as commensalism or mutualism, but under specific conditions some gut microorganisms can become opportunistic parasites or pathogens [4]. For example, insects can directly receive nutrients from microorganisms by digesting them [5]. In addition, microorganisms may increase digestion efficiency or provide nutrients to their host [6] as well as indirectly assist the insect by digesting the diet and extracting nutrients from indigestible materials that thus become available to the insect [7]. Understanding these interactions may lead to the development of strategies to manipulate insect microbiota and boost the mass-rearing process of insects (used, for example, for food and feed purposes or as biological control agents), with undoubted economic benefits.

In this study, we focused on the black soldier fly (BSF, *Hermetia illucens* L.) and its interactions with its gut fungi. BSF larvae represent a sustainable source of protein and fat for animal feed, biodiesel production and other biotechnological applications, as well as suitable agents for organic waste processing and compost/fertilizer production [8–11]. The adult lays eggs on a variety of organic materials and the larva feeds and develops on the decaying matter for at least 13 days [12], although the physical and chemical composition of the substrate can strongly affect the larval development, other life-history traits and the nutritional composition of the larvae [10,13,14]. During their growth, the larvae encounter and consume a variety of microorganisms. Several studies characterized the bacterial community present in the gut of BSF larvae and investigated how this community is affected by the rearing substrates, e.g., [13,15–18]. Furthermore, it has been demonstrated that the addition of specific bacteria to the rearing substrate shortens the life cycle of BSF larvae and leads to an increase in their body weight, thus maximizing their growth potential for applied purposes [16]. However, although several efforts have been spent to characterize the relationships between gut bacteria and BSF larvae and their effects on the insect, there is a lack of knowledge about the contribution of gut fungal microorganisms on the BSF life cycle.

Fungi possess complex metabolic capabilities and can provide nutritional components such as proteins, fatty acids, and vitamins to the host [19]. In previous studies, yeast-like fungus of the genera *Candida* and *Pichia* were found to be dominant in the gut of BSF larvae and in their rearing substrate [20–21]. In addition, supplementation of the diet with *C. tropicalis* resulted in increased body weight of BSF larvae [22]. The relationship between fungi and BSF is completely unknown and more research is required on this topic. It is not clear whether these fungi are digested by the larvae, proliferate within their digestive system and release nutrients and metabolites that are absorbed by the insect, or if they provide nutrients to the larvae by modifying substrate composition. Thus, this study investigated the survival of *C. tropicalis* in the BSF gut to evaluate their ability to colonize this organ or their possible digestion by the insect. The effect of *C. tropicalis* on the metabolic composition and digestion process in the larvae was also analyzed and compared to the metabolic composition of *C. tropicalis* biomass, and rearing substrate.

## Materials and methods

### *C. tropicalis* rearing

*Candida tropicalis*, isolated from BSF gut [22], was stored at −80°C in 50% glycerol (Romical, Israel). For the experiments, the 1 ml from the stored *C. tropicalis* were placed in 10 mL aerobic tubes containing 6 ml of YPD broth (Difco™, Becton Dickinson and Company) with an antibiotic (chloramphenicol, Sigma-Aldrich, USA, 25 ng/L) and maintained at 30°C in a shaking incubator. After 3 h the content of the tubes was diluted to 0.6 L and kept in the same conditions for 72 h. *C. tropicalis* were finally collected by centrifugation.

### Fatty acid composition of *C. tropicalis*

To evaluate the effect of the supplemented fungi in the diet in terms of fatty acid composition in the larvae, lipids were extracted from three replicates of 1 gr of *C. tropicalis* biomass that were collected by centrifugation and freeze-dried (Alpha 2–4 LSCbasic, Christ), using the Sohxlet method with hexane (Carlo Erba, France) [23]. Benzophenone (Sigma-Aldrich, USA) was added as an internal standard. After hydrolization of the fatty acids with $H_2SO_4$, the fatty acid methyl esters (FAME) were analyzed by gas–liquid chromatography (7890A, Agilent Technologies) using a Zebron ZB-FAME column (30 m × 0.25 mm × 0.20 µm; Phenomenex, USA) under the following conditions: injector: 240°C; detector: 260°C; $H_2$ as carrier gas; temperature program: 140°C for 2 min, followed by an increase of 10°C/min to 190°C, then 3°C/min to 260°C. Peaks were identified by comparing their retention times with those of the corresponding standards (Supelco 37 Component FAME Mix, Sigma-Aldrich, USA).

### Experimental setting

BSF eggs were purchased from FreezeM™ (Nachshonim, Israel) and sterilized with three washes of 70% ethanol (Romical, Israel) and 1% NaOCl (Romical, Israel). The sterilized eggs were placed in 170 mL plastic containers with a sterilized (in an autoclave, OT 90L, Nüve, Türkiye) standard diet for Diptera (Gainesville diet containing alfalfa, wheat bran, and corn meal, ratio 3:5:2) [24] in a rearing chamber (28°C, 70% RH, TK 252, Nüve, Türkiye).

The experiment was set with BSF larvae after 3 days from hatching, which were counted and transferred to new vials (170 mL) with 25 g of the standard diet and 50 ml distilled water (75 larvae per vials). Ten replicates were supplemented with *C. tropicalis* (Ct, 4 g), and five replicates remained in the diet without *C. tropicalis*. The larvae were kept on these diets for 10 days: then, five replicates of larvae that were grown on diet with supplemented *C. tropicalis* were transferred to a sterilized substrate (Ct-) for one day, and the other five replicates that were supplemented with *C. tropicalis* remained in the same diet (Ct+), the aim of this change in substrate type was to test whether *C. tropicalis* colonization of the gut decreased after the removal of *C. tropicalis* supplementation from the substrate or whether *C. tropicalis* remained alive in the gut. The hypothesis is that in the absence of colonization of the BSF larval gut by *C. tropicalis*, the fungi will be eliminated from the gut along with the gut contents during the period the larvae spend in a substrate not supplemented with *C. tropicalis* (Ct− treatment). The larvae that were grown in the diet without the supplemented *C. tropicalis* remained in this type of diet throughout the rearing process and served as control (Fig 1).

### Evaluation of the presence of live *C. tropicalis* in the larval gut

To assess the presence of live *C. tropicalis* in the BSF gut, fungal RNA was relatively quantified. For that purpose, three last instar larvae were collected from each treatment and dissected in phosphate-buffered saline (Bio-Lab, Israel). The gut was isolated and separated into five regions (foregut, anterior midgut, middle midgut, posterior midgut, and hindgut) according to the physiological description provided by [25]. To ensure accurate identification of the midgut, the pH of each region was measured using pH indicator strips (Macherey-Nagel, Germany). Each gut district, with the enclosed luminal content, was homogenized with 600 µl RiboEX (GeneAll, Korea) and stored at −80°C until RNA extraction. RNA extraction

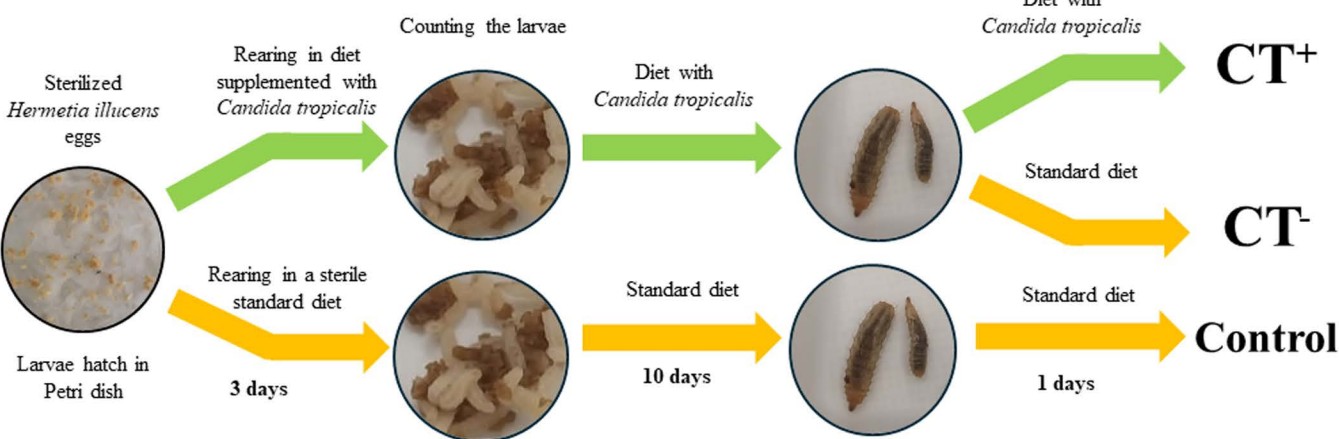

**Fig 1. Experimental design and the different feeding conditions.**

was conducted using the Hybris-R kit (GeneAll, Korea) according to the manufacturer and the obtained RNA was cleaned using a Clean-up RNA concentrator kit (A&A Biotechnology, Poland). cDNA was synthesized using a Verso cDNA synthesis kit (Thermo Fisher Scientific, USA). To evaluate and quantify the presence of live fungi in the gut, two sets of primers were used: R5 (5-ACTTGTTCGCTATCGGTCTC-3) and RF2 (5-CGTCATAGAGGGTGAGAATCC-3) for the 26S fungal rRNA [26], and *Hi*RPL5F (5-AGTCAGTCTTTCCCTCACGA-3) and *Hi*RPL5R (5-GCGTCAACTCGGATGCTA-3) (*Hermetia illucens* ribosomal protein L5) as housekeeping gene [27]. The PCR reaction was conducted in a CFX96 Touch™ Real-Time PCR (BioRad, USA) with 2 µl of HY-SYBR enzyme (Hy-Labs, Israel), 0.3 µl of each primer, and 3 µl of RNA. The amplification protocol included 40 cycles at 95°C for 10 s, 60°C for 10 s, and 72°C for 20 s. The samples were normalized with the expression of the housekeeping gene and quantification was not absolute but relative between the treatments

### Determination of the metabolic composition of the larvae, *C. tropicalis* and rearing substrate using untargeted metabolic analysis

To test whether the larvae consumed the fungi directly or whether *C. tropicalis* affected the larvae indirectly through changes in the substrate, the metabolic composition of the larvae that were fed with and without supplemented *C. tropicalis* was compared. In addition, the metabolic changes in the larvae were also compared to the metabolic composition of *C. tropicalis* biomass and metabolic changes in the feeding substrate in the presence of *C. tropicalis*.

For the untargeted metabolomic analysis, ten larvae from each replicate and five g of substrate were ground with liquid nitrogen into a homogeneous powder. *C. tropicalis* was grown in YPD broth medium and collected by centrifugation at 4,000 rpm at 4°C 4 min. The pellet was collected to Eppendorf tubes in an ice box and sonicated (Qsonica, USA) for 1 min and 15 s with 15 s pulse on and 30 s pulse off to break the cell walls, followed by freeze-drying using a lyophilizer (Alpha 2–4 LSCbasic, CHRIST). One g of each sample was mixed with 5 mL 80% methanol (Carlo Erba, France) and 0.1% formic acid (99% Carlo Ebra, France), vortexed, and centrifuged at 13,000 *x g* at 4°C for 20 min [22]. The supernatant was diluted with distilled water to a final concentration of 53% methanol. The mixture was again centrifuged at 13,000 g and 4°C. The supernatant was filtered using a 0.2 µm membrane filter and (BioFil, China) stored at −20°C until the analysis. Untargeted metabolomic analysis was conducted as described in [23]. Briefly, the samples were analyzed by injecting 5 µl of the extracted solutions into a UHPLC (ultra-high performance liquid chromatography) apparatus connected to a photodiode array detector (Dionex Ultimate 3000), with a reverse-phase column (ZORBAX Eclipse plus C18, 3.0*100 mm, 1.8 µm). Data pre-processing, such as peak determination and peak area integration, were performed with Compound

Discoverer 3.3 (Thermo Xcalibur, UK). Peak areas from each sample were normalized to the quality controls. Annotation of metabolites was based on the similarity scores between the unknown features in the sample and identified compounds in reference databases (MzCloud database using data from the second phase of mass spectrometry (MS2) to obtain accurate qualitative and relative quantitative results & ChemSpider database using high-resolution mass spectrometry). The similarity between fragments of the spectrum (MS2 data) reduced the ambiguities left by simply mass matching using scans from the first stage of mass spectrometry (MS1). Each feature (i.e., a metabolite with a unique m/z and retention time) in our study was detected by mass via high-resolution MS (HRMS), then Compound Discoverer 3.3 was used to match the chemical formula by isotope abundance. Subsequently, the highest score for structure identification was matched from hundreds of possible structures for each chemical formula (from the above database). The obtained chromatogram peaks were detected and peak area integration was performed. The peaks were identified using Compound Discoverer 3.3 equipped with multiple database and spectral library search tools including mzCloud™ and Chemspider™ to obtain accurate qualitative and relative quantitative results.

### Analysis of enzyme activity in BSF gut

To test whether the larvae consumed the supplemented fungi, the digestive enzyme activity in the larval gut was analyzed. The peritrophic matrix with the enclosed intestinal content was isolated from the middle and posterior midgut [25] and centrifuged at 15,000 $x\,g$ at 4°C for 10 min. The supernatant (i.e., midgut juice) was collected and stored at −80°C until use. Total proteolytic activity was measured in the midgut juice from posterior midgut by using 1% azocasein (Sigma-Aldrich, Italy) as a substrate, as reported by [25]. The samples were collected for no more than one hour, to prevent enzyme degradation, and kept on ice throughout the process. The samples were diluted 1:100 with universal buffer and incubated with the substrate for 30 min at 45°C. The reaction was stopped using 300 µl of 12% trichloroacetic acid (Sigma-Aldrich, USA). The samples were transferred to 10°C for 30 min and centrifuged at 15,000 $x\,g$ for 10 min at 4°C. An equal volume of 500 mM NaOH (Sigma-Aldrich, USA) was added. The reaction was measured at 440 nm with a spectrometer (Pharmacia Biotech Ultrospec 3000 UV-Visible, Biochrom Ltd.). Lysozyme activity was measured in the midgut juice from the middle midgut by using lyophilized *Micrococcus lysodeikticus* cells (Sigma-Aldrich) as a substrate [25]. The samples were diluted with 20 µl of $K_3PO_4$ buffer (pH 6.2) and 980 µl of substrate. The reaction was measured at 450 nm with a spectrometer.

### Data analysis

To test for differences in the abundances of live fungi in the gut, data was transformed using log (x + 1), normality was assessed using the Shapiro-Wilk test, and homogeneity of variances was checked using Levene's test. As the ANOVA assumptions were not met, a non-parametric Kruskal Wallis test was performed. Separate analyses were used with the treatments and the gut region as factors to compare the feeding protocols. A comparison of the enzyme activity was conducted using a t-test. Data were analyzed using Jamovi (The Jamovi project, 2022). The LC-MS/MS peak area medians, standard deviation (SD), and relative SD (RSD) were calculated using Compound Discoverer 3.3. Significant differences between groups were considered significant at *p*-value ≤ 0.05. The analysis of metabolic pathways in BSF larvae was performed using MetaboAnalyst 5.0, with a high-quality inbuilt KEGG metabolic pathway database. The KEGG pathway library of yeast (*Saccharomyces cerevisiae*) or insect (*Drosophila melanogaster*) was used for pathway analysis in MetaboAnalyst 5.0.

## Results

### Abundance of live *C. tropicalis* and bacteria in the BSF larval gut

The treatments (i.e., different substrates) and midgut regions showed differences in the abundance of live *C. tropicalis* (Treatments: $X^2_{df2}$=8.01, P = 0.02; Midgut regions: $X^2_{df3}$ = 7.64, P = 0.05; Fig 2). The abundance of live *C. tropicalis* was

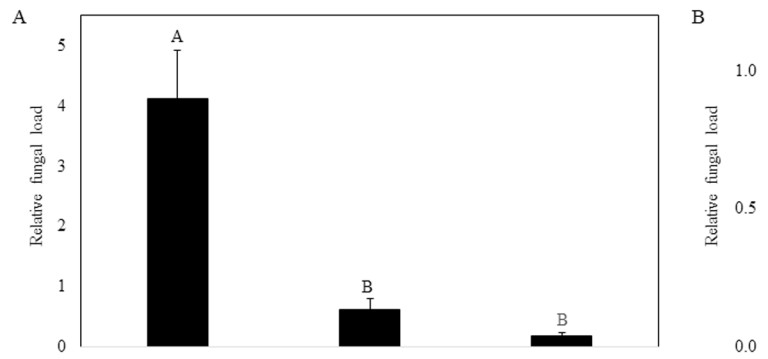 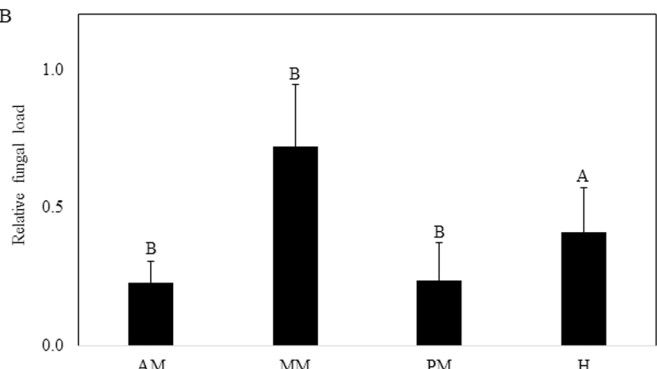

**Fig 2. (A)** Relative *Candida tropicalis* load in the gut of BSF larvae from different treatments: supplemented *C. tropicalis* throughout the rearing process – CT+; supplemented *C. tropicalis* throughout the rearing process except for the last day – CT-; no supplemented *C. tropicalis* – control). **(B)** relative *C. tropicalis* load in different gut regions (anterior midgut – AM, middle midgut – MM, posterior midgut – PM, and hindgut – H). The values represent averages with standard deviation of gene 26S expression. Significant differences (p<0.05) are marked by different letters above the bars and error bars represent standard error.

higher in the treatments in which the larvae were constantly fed with *C. tropicalis* (Ct+ treatment) compared to the treatment where *C. tropicalis* supplementation ceased after five days (Ct-). A comparable amount of *C. tropicalis* derived from Ct- was detected in the control, where *C. tropicalis* was not supplemented to the substrate (Fig 2A). In addition, the highest abundance of live *C. tropicalis* in the gut was found in the middle midgut (MM) (Fig 2B). The foregut was excluded from the analyses due to low RNA content.

### Fatty acid composition of *C. tropicalis*

The *C. tropicalis* sample was composed of the main fatty acids: lauric acid (C12:0, 66.4% ± 1.95; average ± standard error), palmitic acid (C16:0, 19.8% ± 0.7), myristic acid (C15:0, 6.7% ± 1.3), and linoleic acid (C18:2, 2% ± 1.7) (Fig 3).

### Metabolic composition of larvae, fungi, and rearing substrate

The metabolic pathways that increased in the larvae that consumed *C. tropicalis* were fatty acid biosynthesis (2 metabolites out of 42), purine metabolism (2/65), and valine, leucine, and isoleucine biosynthesis (1/8) (Table 1), while pathways that increased in *C. tropicalis* biomass relative to other fungi (*Saccharomyces cerevisiae*) were biosynthesis of unsaturated fatty acids (4/23) and tyrosine metabolism (2/15) (Table 2).

The rearing substrate that included *C. tropicalis* showed an increase in metabolites from the following pathways: tyrosine metabolism (12/42), arachidonic acid metabolism (12/36), valine leucine and isoleucine biosynthesis (6/8), arginine and proline metabolism (10/38), linoleic acid metabolism (3/5), valine leucine and isoleucine degradation (8/40), pentose and glucuronate interconversion (5/18), arginine biosynthesis (4/14), tryptophan metabolism (7/41), vitamin B6 metabolism (3/9), histidine metabolism (4/16), D-Arginine and D-ornithine metabolism (2/4), and phenylalanine metabolism (3/10) (Table 3).

### Enzyme activity

Lysozyme activity was lower in larvae that were fed on standard diet supplemented with *C. tropicalis* compared to larvae reared on a standard diet without supplementation (Control) ($t_{df2}$=4.6, P=0.04; Fig 4A); however, the activity of proteolytic enzymes did not differ between larvae that were fed on diet supplemented with *C. tropicalis* and larvae fed on control diet ($t_{df7}$=0.78, P=0.46; Fig 4B).

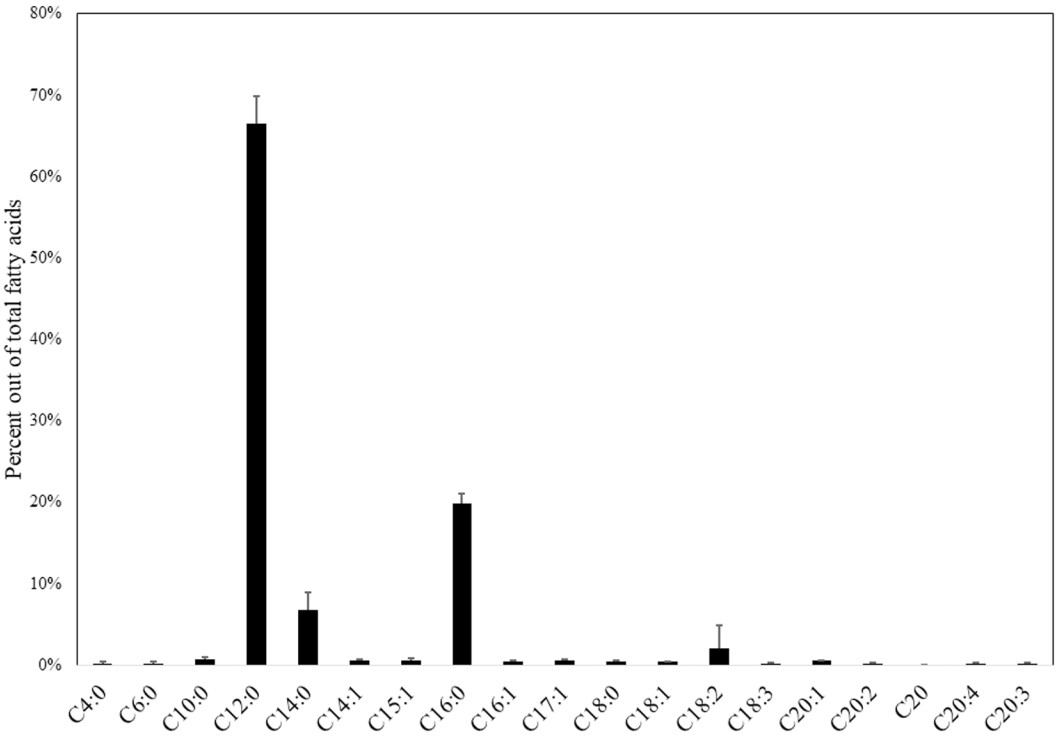

**Fig 3. Percent (in weight) of the different fatty acids (represented by the numerical symbol of the total amount of carbon atoms of the fatty acid and the number of double bonds) related to total fatty acids in _Candida tropicalis_.** Error bars represent standard error.

**Table 1. Metabolites and pathways that increased in BSF larvae fed with _Candida tropicalis_ compared to larvae without any supplemented fungi in the substrate.**

| Metabolic pathway | Metabolite names | Matched metabolites | P value |
|---|---|---|---|
| Fatty acid biosynthesis | palmitic acid; myristic acid | 2/47 | 0.01 |
| Purine metabolism | hypoxanthine; guanine | 2/65 | 0.01 |
| Valine, leucine and isoleucine biosynthesis | L-threonine | 1/8 | 0.02 |

**Table 2. Metabolites and pathways that increased in the biomass of _Candida tropicalis_ relative to other fungi (_Saccharomyces cerevisiae_).**

| Metabolic pathway | Metabolite name | Matched metabolites | P value |
|---|---|---|---|
| Biosynthesis of unsaturated fatty acids | linoleate; dihomo-gamma-linolenate; (6Z,9Z,12Z)-octadecatrienoic acid; (9Z,12Z,15Z)-octadecatrienoic acid | 4/23 | <0.01 |
| Tyrosine metabolism | 3-methoxy-4-hydroxyphenylglycolaldehyde; homovanillate | 2/15 | 0.05 |

## Discussion

This study examined the survival of _C. tropicalis_ in the BSF gut and its effect on the metabolic and digestive processes of BSF larvae. Although _C. tropicalis_ was found to be dominant in the BSF larval gut [21], the presence of the live _C. tropicalis_ was reduced in the insect gut when it was removed from the rearing substrate. Surprisingly, the highest abundance

**Table 3. Metabolites and pathways that increased in the BSF rearing substrate supplemented with *Candida tropicalis* compared to substrate without fungi.**

| Metabolic pathway | Metabolite name | Matched metabolites | P value |
|---|---|---|---|
| Tyrosine metabolism | Normetanephrine; 3-Methoxy-4-hydroxyphenylglycolaldehyde; Norepinephrine; Epinephrine; 3,4-Dihydroxymandelic acid; Homovanillin; 4-Hydroxyphenylacetaldehyde; Homovanillic acid; Fumaric acid; Pyruvicacid; Acetoacetic acid; Dopaquinone; Gentisic acid | 13/42 | <0.01 |
| Arachidonic acid metabolism | 5,6-Epoxy-8,11,14-eicosatrienoic acid; 8,9-Epoxyeicosatrienoic acid; 11,12-Epoxyeicosatrienoic acid; Arachidonic acid; 5,6-DHET; 8,9-DiHETrE; 11,12-DiHETrE; 14,15-DiHETrE; 20-Hydroxyeicosatetraenoic acid; 15(S)-HETE; 19(S)-HETE; 5-HETE | 12/36 | <0.01 |
| Valine, leucine and isoleucine biosynthesis | 3-Methyl-2-oxovaleric acid; L-Leucine; Alpha-ketoisovaleric acid; 2-Ketobutyric acid; L-Isoleucine; Ketoleucine; | 6/8 | <0.01 |
| Phenylalanine metabolism | Phenylacetaldehyde; Phenylacetic acid; Hippuric acid; | 3/10 | <0.01 |
| Arginine and proline metabolism | Trans-3-hydroxy-L-proline; D-Proline; 4-Hydroxyproline; L-Proline; cis-4-Hydroxy-D-proline; 1-Pyrroline-5-carboxylic acid; L-Glutamic gamma-semialdehyde; Ornithine; 1-Pyrroline-2-carboxylic acid; Pyruvic acid | 10/38 | <0.01 |
| Linoleic acid metabolism | Linoleic acid; 9,10-Epoxyoctadecenoic acid; 12,13-EpOME | 3/5 | <0.01 |
| Valine, leucine and isoleucine degradation | Acetoacetic acid; Alpha-ketoisovaleric acid; L-Isoleucine; (S)-Methylmalonic acid semialdehyde; 2-Methyl-3-oxopropanoic acid; Ketoleucine; L-Leucine; | 8/40 | <0.01 |
| Pentose and glucuronate interconversions | D-Ribitol 5-phosphate; D-XyluloseD-Xylose; L-Arabinose; L-Threo-2-pentulose | 5/18 | <0.01 |
| Arginine biosynthesis | Citrulline;Ornithine; N-Acetylglutamic acid; Fumaric acid | 4/14 | 0.01 |
| Tryptophan metabolism | 5-Hydroxyindoleacetic acid; 5-Hydroxyindoleacetaldehyde;; Indolepyruvate; 4-(2-Amino-3-hydroxyphenyl)-2,4-dioxobutanoic acid; 4-(2-Aminophenyl)-2,4-dioxobutanoic acid; Indoleacetic acid | 7/41 | 0.01 |
| Vitamin B6 metabolism | Pyridoxine;Pyridoxal; 4-Pyridoxic acid | 3/9 | 0.02 |
| Histidine metabolism | Urocanic acid; L-Histidine; Imidazoleacetic acid; Imidazole-4-acetaldehyde | 4/16 | 0.02 |
| D-Arginine and D-ornithine metabolism | D-Ornithine; 5-Amino-2-oxopentanoic acid; | 2/4 | 0.02 |

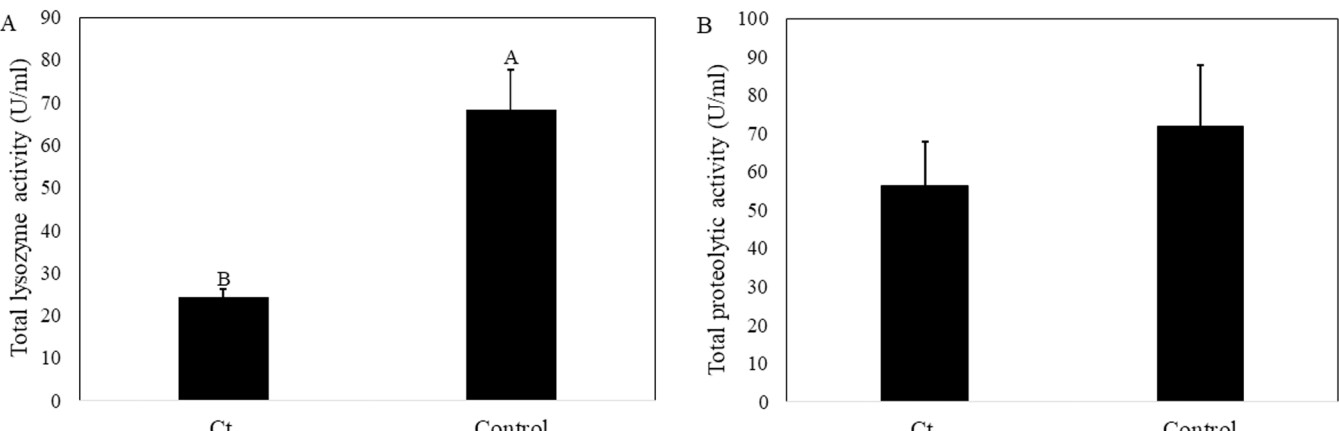

**Fig 4. Lysozyme (A) and total proteolytic (B) activity in the gut of BSF larvae fed with supplemented *Candida tropicalis* (Ct) or without supplemented *C. tropicalis* (Control).** Significant differences ($p < 0.05$) are marked by different letters above the bars; error bars represent standard error.

of live *C. tropicalis* was found in the middle region of the midgut, where the lumen is very acidic (i.e., pH 2; [25]). These findings are novel and suggest that *C. tropicalis* survives the acidic chamber in the middle midgut but do not colonize the gut. These results also suggest that the high abundance of *C. tropicalis* in the BSF gut found in [21] was due to its high abundance in the rearing substrate, maybe caused by a selection process of these fungi in the BSF environment, e.g., due to change in the pH levels.

The increase in the BSF larval weight after supplementation of *C. tropicalis* to the substrate reported in [22] may be due to direct consumption of *C. tropicalis* by the larvae. The apparent capability of BSF larvae to consume *C. tropicalis* is corroborated by the metabolomic analysis. Indeed, the fatty acid biosynthesis pathway increased due to an increase in the palmitic and myristic acids that are abundant in *C. tropicalis* (19.8% ± 0.66 and 6.69% ± 1.31, respectively). However, the high protein content of *C. tropicalis* [22] combined with the lack of increased proteolytic activity in the larvae supplemented with *C. tropicalis* in the substrate may suggest that *C. tropicalis* are not directly consumed.

On the other hand, the effect of the fungi could be indirect due to the consumption by the larvae of metabolites secreted by *C. tropicalis* into the substrate. To support this hypothesis, an increase in metabolites from the valine, leucine, and isoleucine biosynthesis pathways both in the larvae and the rearing substrate was detected. Additionally, an increase in L-threonine was detected in the larvae and an enhancement in acetoacetic acid, alpha-ketoisovaleric acid, L-isoleucine, (S)-methylmalonic acid semialdehyde, 2-methyl-3-oxopropanoic acid, ketoleucine, and L-leucine was registered in the rearing substrate. The leucine and isoleucine amino acids are known to activate the Target of Rapamycin (TOR) pathway that increases protein synthesis and reproduction in insects [28] and could contribute to augmenting the insect body weight. Furthermore, in the substrate containing *C. tropicalis* an increase in metabolites from the tyrosine, vitamin B6, and histidine metabolism pathways was identified. Although tyrosine and histidine are non-essential amino-acid and can be produced by the insect, supplements of these amino acids in the diet can be beneficial to the insect, e,g, tyrosine for the cuticle tanning process [29]. Vitamin B is an essential vitamin that acts as co-factors for different enzymes and insects can acquire this vitamin from the diet or various microorganisms, such as fungi [30]. A similar pattern was observed in a previous study in which BSF larvae were fed with *C. tropicalis* [22]. All these findings suggest that larvae take advantage of fungal-derived metabolites produced in the rearing substrate.

Interestingly, lysozyme activity was reduced in the presence of *C. tropicalis* in the BSF gut. Lysozyme activity can serve as anti-bacterial agent in the insect gut [31] and can be used by detritivores dipterans to digest bacteria and use them as a food source [32]. In some cases, lysozyme can effectively degrade fungal wall [33] and reduce their population through cationic activity [34], therefore can protect the insect from fungal infections in the gut. Although this reduction in lysozyme activity may suggest that *C. tropicalis* influences the insect's immune response, it is important to note that the evidence is primarily correlative at this stage. It may be that *C. tropicalis* can reduce the enzymatic activity of the insect to allow fungal survival. Such an effect was described concerning the immune response of BSF, where *C. tropicalis* caused a reduction of antimicrobial peptide production [35]. However, further experimental work is needed to confirm the exact mechanisms behind these observations. In addition, the presence of *C. tropicalis* may have decreased the bacterial community and therefore led to a reduced need for lysozyme activity, which could explain the lower activity observed in larvae fed with *C. tropicalis* compared to the control.

This study demonstrates that 1) *C. tropicalis* survives the BSF gut but does not colonize it; 2) two potential mechanisms can explain the nutritional effect of *C. tropicalis* on BSF larvae. The first involves direct digestion of *C. tropicalis* by the larvae, while the second implies an indirect effect of this species due to the extraction of nutrients into the substrate, which is then consumed by the larva; and 3) *C. tropicalis* can potentially manipulate the production and activity of digestive enzyme, which may be harmful to fungal survival in the larval gut.

These findings, together with the high abundance of *C. tropicalis* in the larval natural environment found in other studies, suggest indirect interactions. In these interactions, *C. tropicalis* thrives in the surrounding of the BSF larvae, due to special conditions, e.g. pH level, and may manipulates the BSF digestive enzyme activity to survive in this environment as

it passes through the insects' digestive tract. However, the BSF larvae benefit indirectly from *C. tropicalis* presence in the environment due to the secretion of metabolites by *C. tropicalis*. Despite these interesting insights, further experimental validation is required to establish these mechanisms with certainty. These complex interactions need to be further investigated for a full characterization.

## Author contributions

**Conceptualization:** Lilach Ben-Mordechai, Tzach Vitenberg, Gianluca Tettamanti, Morena Casartelli, Daniele Bruno, Itai Opatovsky.

**Data curation:** Lilach Ben-Mordechai, Neta Herman, Tzach Vitenberg, Morena Casartelli, Daniele Bruno.

**Formal analysis:** Lilach Ben-Mordechai, Sivan Margalit.

**Funding acquisition:** Itai Opatovsky.

**Investigation:** Lilach Ben-Mordechai.

**Methodology:** Lilach Ben-Mordechai, Neta Herman, Tzach Vitenberg, Gianluca Tettamanti, Morena Casartelli, Daniele Bruno, Itai Opatovsky.

**Supervision:** Itai Opatovsky.

**Visualization:** Lilach Ben-Mordechai.

**Writing – original draft:** Lilach Ben-Mordechai.

**Writing – review & editing:** Lilach Ben-Mordechai, Neta Herman, Sivan Margalit, Gianluca Tettamanti, Morena Casartelli, Daniele Bruno, Itai Opatovsky.

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
