## [Decision Letter · Decision Letter 0]

Dear Dr. Opatovsky,

Thank you for submitting your manuscript to PLOS ONE. After careful consideration, we feel that it has merit but does not fully meet PLOS ONE’s publication criteria as it currently stands. Therefore, we invite you to submit a revised version of the manuscript that addresses the points raised during the review process.

We look forward to receiving your revised manuscript.

Kind regards,

Andrey Nagdalian

Academic Editor

PLOS ONE

 Journal requirements: When submitting your revision, we need you to address these additional requirements. 1. Please ensure that your manuscript meets PLOS ONE's style requirements, including those for file naming. The PLOS ONE style templates can be found at https://journals.plos.org/plosone/s/file?id=wjVg/PLOSOne_formatting_sample_main_body.pdf and https://journals.plos.org/plosone/s/file?id=ba62/PLOSOne_formatting_sample_title_authors_affiliations.pdf. 2. We suggest you thoroughly copyedit your manuscript for language usage, spelling, and grammar. If you do not know anyone who can help you do this, you may wish to consider employing a professional scientific editing service.  The American Journal Experts (AJE) (https://www.aje.com/) is one such service that has extensive experience helping authors meet PLOS guidelines and can provide language editing, translation, manuscript formatting, and figure formatting to ensure your manuscript meets our submission guidelines. Please note that having the manuscript copyedited by AJE or any other editing services does not guarantee selection for peer review or acceptance for publication.  Upon resubmission, please provide the following: The name of the colleague or the details of the professional service that edited your manuscript A copy of your manuscript showing your changes by either highlighting them or using track changes (uploaded as a *supporting information* file) A clean copy of the edited manuscript (uploaded as the new *manuscript* file)” 3. Thank you for stating the following financial disclosure:  [Israel Science Foundation (grant No. 1167/21)].  Please state what role the funders took in the study.  If the funders had no role, please state: ""The funders had no role in study design, data collection and analysis, decision to publish, or preparation of the manuscript."" If this statement is not correct you must amend it as needed. Please include this amended Role of Funder statement in your cover letter; we will change the online submission form on your behalf. 4. We note that you have indicated that there are restrictions to data sharing for this study. PLOS only allows data to be available upon request if there are legal or ethical restrictions on sharing data publicly. For more information on unacceptable data access restrictions, please see http://journals.plos.org/plosone/s/data-availability#loc-unacceptable-data-access-restrictions.  Before we proceed with your manuscript, please address the following prompts: a) If there are ethical or legal restrictions on sharing a de-identified data set, please explain them in detail (e.g., data contain potentially identifying or sensitive patient information, data are owned by a third-party organization, etc.) and who has imposed them (e.g., a Research Ethics Committee or Institutional Review Board, etc.). Please also provide contact information for a data access committee, ethics committee, or other institutional body to which data requests may be sent. b) If there are no restrictions, please upload the minimal anonymized data set necessary to replicate your study findings to a stable, public repository and provide us with the relevant URLs, DOIs, or accession numbers. For a list of recommended repositories, please seehttps://journals.plos.org/plosone/s/recommended-repositories. You also have the option of uploading the data as Supporting Information files, but we would recommend depositing data directly to a data repository if possible. We will update your Data Availability statement on your behalf to reflect the information you provide. 5. In the online submission form, you indicated that [data available on request from the author]. All PLOS journals now require all data underlying the findings described in their manuscript to be freely available to other researchers, either 1. In a public repository, 2. Within the manuscript itself, or 3. Uploaded as supplementary information.This policy applies to all data except where public deposition would breach compliance with the protocol approved by your research ethics board. If your data cannot be made publicly available for ethical or legal reasons (e.g., public availability would compromise patient privacy), please explain your reasons on resubmission and your exemption request will be escalated for approval. 

Reviewers' comments:

Reviewer's Responses to Questions

**Comments to the Author**

1. Is the manuscript technically sound, and do the data support the conclusions?

Reviewer #1: Partly

Reviewer #2: Yes

2. Has the statistical analysis been performed appropriately and rigorously?

Reviewer #1: No

Reviewer #2: Yes

3. Have the authors made all data underlying the findings in their manuscript fully available?

Reviewer #1: Yes

Reviewer #2: No

4. Is the manuscript presented in an intelligible fashion and written in standard English?

Reviewer #1: Yes

Reviewer #2: Yes

Reviewer #1: Questions

The manuscript deals with an interesting topic concerning Candida tropicalis found in the digestive tract of Hermetia illucens. This study presents interesting findings. However, there are several major concerns that need to be addressed.

Here's a list of potential limitations or areas for further investigation

1. The title uses the term "non-orthodox symbiotic" without providing a clear definition in the article. It is a highly specific term that requires a precise explanation and data confirmation.

2. There is inconsistent use of professional terminology throughout the article, such as fungi, yeast, and C. tropicalis. And the terminology for black soldier fly. The article uses "black soldier fly," "BSF," and "Hermetia illucens." Although these are all correct, the full name and abbreviation should be given at first mention, and then one term should be used consistently.

3. This article is based on the study by Kannan et al. (2023). The authors used different C. tropicalis feeding strategies for BSF larvae. However, was there a significant increase in BSF body weight after supplementing their diet with C. tropicalis? Could the authors provide supporting evidence for this effect?

4. Line 214: The meaning of significant differences between A and B was not mentioned in the article

5. Line 209-215: There is a lack of a unit for the y-axis in Figure 2.

6. How many BSF were used in Figure 2a and Figure 2b.

The article doesn't clearly state the sample size and number of replications for each experiment. Small sample sizes could affect the reliability and generalizability of results.

7. How did the authors confirm that the separated gut sections were correctly identified as the anterior midgut, middle midgut, and posterior midgut? Did the authors have anatomical data or specific markers for AM, MM, and PM to validate these gut sections?

8. Line 216-221: The methodology for Figure 3 was not described in the Materials and Methods section. Please provide information about the source and purity of standards used for quantification.

The fatty acid abbreviations on the x-axis in Figure 3 lack detailed explanations. Please clarify whether the y-axis represents weight percentage or molar percentage of total fatty acids.

9. Line 285-289: The conclusions drawn by Genta et al. (2009) appear to contradict the arguments presented by the authors in this paper. This discrepancy needs to be addressed and explained.

10. Line 287: Please demonstrate the relationship between lysozyme activity and the survival rate of BSF larvae.

11. The authors should clarify whether C. tropicalis enhances or suppresses the immune response in BSF larvae. Additionally, several genes associated with BSF survival have been previously identified. Could you examine and discuss the potential relationship between these known survival-related genes and the immune response observed in the discussion?

Reviewer #2: The authors studied investigated the survival of C. tropicalis in the BSF gut to evaluate their ability to colonize this organ or their possible digestion by the insect. The effect of C. tropicalis on the metabolic composition and digestion process in the larvae was also analyzed and compared to the metabolic composition of the fungal biomass, and rearing substrate. Generally, the conceptualization and results of this work are interesting and the manuscript deserves consideration. However, at presented state the manuscript needs revision. The main comments and recommendations are listed below.

Nutritional composition of BSFL Is changeable and can be regulated using different feed stuff. A little discussion on this aspect could improve the Introduction section. For example: doi.org/10.3920/JIFF2021.0162, 10.1093/ee/nvab135

Details for equipment and software used in the experiment should be unified: the name of manufacturer and its location should be given.

Add separate subsection in section 2 with chemicals and materials. Provide information on chemicals purity and manufacturers.

Tables can be modified. It is better to add separate columns with data on investigated values for experimental and control groups of BSF larvae.

As suggestion, all figures can be combined to the one figure. In this regard, it will be visually comfortable for readers to compare the main results obtained.

The text should be carefully checked for typos and grammatical errors.

**Do you want your identity to be public for this peer review?** For information about this choice, including consent withdrawal, please see our Privacy Policy

Reviewer #1: No

Reviewer #2: No

---

## [Author Response · Author response to Decision Letter 1]

16 Feb 2025

Reviewer #1: Questions

The manuscript deals with an interesting topic concerning Candida tropicalis found in the digestive tract of Hermetia illucens. This study presents interesting findings. However, there are several major concerns that need to be addressed.

Here's a list of potential limitations or areas for further investigation

1. The title uses the term "non-orthodox symbiotic" without providing a clear definition in the article. It is a highly specific term that requires a precise explanation and data confirmation.

We removed this term from the title and discussion and changed it to “indirect interactions” as the effect of the C. tropicalis on the BSF seems to be through the substrate

2. There is inconsistent use of professional terminology throughout the article, such as fungi, yeast, and C. tropicalis. And the terminology for black soldier fly. The article uses "black soldier fly," "BSF," and "Hermetia illucens." Although these are all correct, the full name and abbreviation should be given at first mention, and then one term should be used consistently.

We agree, we improved the term’s consistency in the manuscript by refereeing to C. tropicalis and BSF

3. This article is based on the study by Kannan et al. (2023). The authors used different C. tropicalis feeding strategies for BSF larvae. However, was there a significant increase in BSF body weight after supplementing their diet with C. tropicalis? Could the authors provide supporting evidence for this effect?

Unfortunately, we did not measure the larval body mass in this experiment

4. Line 214: The meaning of significant differences between A and B was not mentioned in the article

The meaning of significance was added to the figure legends

5. Line 209-215: There is a lack of a unit for the y-axis in Figure 2.

The amount of DNA was compared to a relative sample, therefore it is not quantification but relative measurement.

6. How many BSF were used in Figure 2a and Figure 2b.

The article doesn't clearly state the sample size and number of replications for each experiment. Small sample sizes could affect the reliability and generalizability of results.

The sample sizes were added throughout the manuscript

7. How did the authors confirm that the separated gut sections were correctly identified as the anterior midgut, middle midgut, and posterior midgut? Did the authors have anatomical data or specific markers for AM, MM, and PM to validate these gut sections?

The gut parts were separated according to the physiological description provided by Bonelli et al. (2019) and were ensured accurate identification of the midgut, using the pH levels known to maintained in each region (lines 128-131)

8. Line 216-221: The methodology for Figure 3 was not described in the Materials and Methods section.

The methodology explained in lines 93-104

Please provide information about the source and purity of standards used for quantification.

The relevant information was added

The fatty acid abbreviations on the x-axis in Figure 3 lack detailed explanations. Please clarify whether the y-axis represents weight percentage or molar percentage of total fatty acids.

The figure axis was changed accordingly

9. Line 285-289: The conclusions drawn by Genta et al. (2009) appear to contradict the arguments presented by the authors in this paper. This discrepancy needs to be addressed and explained.

If the lysozyme afect the fungal survival, we suggest that C. tropicalis can reduce the enzymatic activity of the insect to allow fungal survival. It is describes in lines 305-308

10. Line 287: Please demonstrate the relationship between lysozyme activity and the survival rate of BSF larvae.

We meant the survival of the fungi (corrected in the text)

11. The authors should clarify whether C. tropicalis enhances or suppresses the immune response in BSF larvae.

The effect of Candida on the insect immune system is specific in line 307

Additionally, several genes associated with BSF survival have been previously identified. Could you examine and discuss the potential relationship between these known survival-related genes and the immune response observed in the discussion?

It is not clear to us which genes that related to the survival of the BSF the reviewer suggests. The effect of the effect of fungi on the immune response in discussed in the paper by Herman et al. 2024.

Reviewer #2: The authors studied investigated the survival of C. tropicalis in the BSF gut to evaluate their ability to colonize this organ or their possible digestion by the insect. The effect of C. tropicalis on the metabolic composition and digestion process in the larvae was also analyzed and compared to the metabolic composition of the fungal biomass, and rearing substrate. Generally, the conceptualization and results of this work are interesting and the manuscript deserves consideration. However, at presented state the manuscript needs revision. The main comments and recommendations are listed below.

Nutritional composition of BSFL Is changeable and can be regulated using different feed stuff. A little discussion on this aspect could improve the Introduction section. For example: doi.org/10.3920/JIFF2021.0162, 10.1093/ee/nvab135

These references were added and the topic is discussed in the introduction (lines 57-59)

Details for equipment and software used in the experiment should be unified: the name of manufacturer and its location should be given.

Add separate subsection in section 2 with chemicals and materials. Provide information on chemicals purity and manufacturers.

The information was added (not in a separate section but in the text)

Tables can be modified. It is better to add separate columns with data on investigated values for experimental and control groups of BSF larvae.

As the information of the LCMS contains a vast number of metabolites, usually the analysis is conducted as a comparison between the treatments and the control. Therefore the tables present only the significant difference in these comparisons

As suggestion, all figures can be combined to the one figure. In this regard, it will be visually comfortable for readers to compare the main results obtained.

As the figures present different analyses of different treatments we think it will be more distracting to combine them.

The text should be carefully checked for typos and grammatical errors.

The text was checked thoroughly

---

## [Decision Letter · Decision Letter 1]

Dear Dr. Opatovsky,

Thank you for submitting your manuscript to PLOS ONE. After careful consideration, we feel that it has merit but does not fully meet PLOS ONE’s publication criteria as it currently stands. Therefore, we invite you to submit a revised version of the manuscript that addresses the points raised during the review process.

We look forward to receiving your revised manuscript.

Kind regards,

Andrey Nagdalian

Academic Editor

PLOS ONE

Reviewers' comments:

Reviewer's Responses to Questions

**Comments to the Author**

Reviewer #1: All comments have been addressed

Reviewer #2: All comments have been addressed

2. Is the manuscript technically sound, and do the data support the conclusions?

Reviewer #1: Yes

Reviewer #2: Yes

3. Has the statistical analysis been performed appropriately and rigorously?

Reviewer #1: No

Reviewer #2: Yes

4. Have the authors made all data underlying the findings in their manuscript fully available?

Reviewer #1: Yes

Reviewer #2: Yes

5. Is the manuscript presented in an intelligible fashion and written in standard English?

Reviewer #1: Yes

Reviewer #2: Yes

Reviewer #1: This manuscript provides an interesting and valuable study on the relationship between the fungus C. tropicalis and BSF. The work makes a meaningful contribution to understanding insect-microbe interactions in an economically important insect species. However, there are several points that could be improved before publication:

Methodology should be clear

1. The experimental design is somewhat complex with different treatments. While Fig 1, the text could more clearly explain the rationale behind the 10-day with C. tropicalis followed by 1-day without design. What was the hypothesis behind this specific timing?

2. The authors use the 26S rRNA gene to assess the presence of C. tropicalis in the gut, but this approach may require further clarification:

1) Authors should consider using absolute quantitative PCR to establish standard curves for more accurate quantification of fungal amounts.

2) Additional methods, such as culture counts or microscopic observation, may be needed to validate the results.

3) Therefore, providing more details on how C. tropicalis was quantified would strengthen the methods section.

3. The manuscript would benefit address more information about how data normality was assessed before choosing between parametric and non-parametric tests.

The evidence is insufficient to support what this study demonstrates.

4. The paper suggests C. tropicalis manipulates BSF digestive enzyme production, but the evidence is primarily correlative. Authors should either provide stronger evidence for this claim or acknowledge its speculative nature more explicitly.

5. The research data in this study is insufficient to support conclusions. The limited dataset does not provide adequate evidence to make such definitive claims about the relationship between C. tropicalis and BSF larvae. More comprehensive investigations would be needed to establish these mechanisms with certainty.

Reviewer #2: The authors considered all comments and recommendations. The revised manuscript deserves further consideration by Editorial board.

**Do you want your identity to be public for this peer review?** For information about this choice, including consent withdrawal, please see our Privacy Policy

Reviewer #1: No

Reviewer #2: No

---

## [Author Response · Author response to Decision Letter 2]

1 Apr 2025

1. The experimental design is somewhat complex with different treatments. While Fig 1, the text could more clearly explain the rationale behind the 10-day with C. tropicalis followed by 1-day without design. What was the hypothesis behind this specific timing?

# An additional explanation was added to the method section (lines 114-116)

2. The authors use the 26S rRNA gene to assess the presence of C. tropicalis in the gut, but this approach may require further clarification:

1) Authors should consider using absolute quantitative PCR to establish standard curves for more accurate quantification of fungal amounts.

2) Additional methods, such as culture counts or microscopic observation, may be needed to validate the results.

3) Therefore, providing more details on how C. tropicalis was quantified would strengthen the methods section.

# We conducted relative quantification and the method was better described (lines 139-141). Unfortunately, we did not conduct any additional methods to support the results.

3. The manuscript would benefit address more information about how data normality was assessed before choosing between parametric and non-parametric tests.

# The relevant data was included (lines 199-201)

The evidence is insufficient to support what this study demonstrates.

4. The paper suggests C. tropicalis manipulates BSF digestive enzyme production, but the evidence is primarily correlative. Authors should either provide stronger evidence for this claim or acknowledge its speculative nature more explicitly.

# The paragraph was changed accordingly (lines 296-309)

5. The research data in this study is insufficient to support conclusions. The limited dataset does not provide adequate evidence to make such definitive claims about the relationship between C. tropicalis and BSF larvae. More comprehensive investigations would be needed to establish these mechanisms with certainty.

# A disclaimer was added in the discussion (lines 322-323)

---

## [Decision Letter · Decision Letter 2]

The fate of Candida tropicalis in the black soldier fly larvae and its nutritional effect suggest indirect interactions

PONE-D-24-31724R2

Dear Dr. Opatovsky,

We’re pleased to inform you that your manuscript has been judged scientifically suitable for publication and will be formally accepted for publication once it meets all outstanding technical requirements.

Kind regards,

Andrey Nagdalian

Academic Editor

PLOS ONE

Additional Editor Comments (optional):

Reviewers' comments:

Reviewer's Responses to Questions

**Comments to the Author**

Reviewer #2: All comments have been addressed

2. Is the manuscript technically sound, and do the data support the conclusions?

Reviewer #2: Yes

3. Has the statistical analysis been performed appropriately and rigorously?

Reviewer #2: Yes

4. Have the authors made all data underlying the findings in their manuscript fully available?

Reviewer #2: Yes

5. Is the manuscript presented in an intelligible fashion and written in standard English?

Reviewer #2: Yes

Reviewer #2: The authors considered all comments and recommendations and decided them well. The revised manuscript deserves acceptance

**Do you want your identity to be public for this peer review?** For information about this choice, including consent withdrawal, please see our Privacy Policy

Reviewer #2: No

---

## [Editor Report · Acceptance letter]

PONE-D-24-31724R2

PLOS ONE

Dear Dr. Opatovsky,

I'm pleased to inform you that your manuscript has been deemed suitable for publication in PLOS ONE. Congratulations! Your manuscript is now being handed over to our production team.

Kind regards,

on behalf of

Dr. Andrey Nagdalian

Academic Editor

PLOS ONE